# Evaluating Green Innovation Efficiency and Its Socioeconomic Factors Using a Slack-Based Measure with Environmental Undesirable Outputs

**DOI:** 10.3390/ijerph182412880

**Published:** 2021-12-07

**Authors:** Hongxu Guo, Zihan Xie, Rong Wu

**Affiliations:** 1School of Architecture and Urban Planning, Guangdong University of Technology, 729 East Dongfeng Road, Guangzhou 510090, China; guohx@163.com; 2Guangdong Provincial Key Laboratory of Urbanization and Geo-simulation, School of Geography and Planning, Sun Yat-sen University, Guangzhou 510275, China

**Keywords:** green innovation efficiency, the Pearl River Delta, Super-SBM model, Tobit model, Global Malmquist–Luenberger index

## Abstract

Understanding green innovation efficiency (GIE) is crucial in assessing achievements of the current development strategy scientifically. Existing literature on measuring green innovation efficiency with considering environmental undesirable outputs at the city level is limited. Consulting existing studies, this paper constructs an evaluation index system to measure green innovation efficiency and its socioeconomic impact factors. Employing a super slacks-based measure (Super-SBM) model, which takes into account undesirable outputs (industrial wastewater emissions, industrial exhaust emissions and CO_2_ emissions), and a Global Malmquist–Luenberger index (GML), we calculate the green innovation efficiency of 15 cities in the Pearl River Delta (PRD) urban agglomeration from 2009 to 2017, exploring the impact factors behind green innovation efficiency using a Tobit panel regression model. The empirical results are as follows: Due to the heterogeneity of urban functional division and economic development in the Pearl River Delta, more than half of the region’s cities were found to be in ineffective or transitional states with respect to their green innovation efficiency. A GML decomposition index shows that technological efficiency and technological progress are out of step with one another in the Pearl River Delta, an asymmetry which is restricting regional green innovation growth. The influencing factors of industrial structure, the level of economic openness, and the urban informationization level are shown to have promoted green innovation efficiency in the Pearl River Delta’s cities, while government R&D expenditure and education expenditure exerted negative effects. This paper concludes by highlighting the importance of cooperation between the government and enterprises in achieving green innovation.

## 1. Introduction

Following the economic system reform, Chinese society entered a phase of rapid development and structural transformation. This growth came at the cost of a series of persistent environmental issues, which have become apparent in recent decades [1] and have hindered China’s regions in shrugging off path dependencies and updating outmoded roles [2]. The construction of market economy system has stimulated the activity and diversification of economic behaviors in China. Promoted by rapid increases in the wealth of its society and continuous improvements in residents’ living standards, the structure and level of consumption are also optimized [3]. Consequently, China has now stepped into the “New Normal” period, as late industrialization has given way to a post-industrial economic development model. With the liberation of social productivity and increases in income levels, China’s economy is characterized by a cyclic loop of “high savings and high consumption” [4]. The country’s high savings rate has accelerated urbanization and economic development in recent decades [5], with the highest rates occurring during the 20- to 25-year period following China’s economic takeoff. A decreasing savings rate is thus an inevitable challenge that will need to be faced in the future development of the country, and it will be necessary for China to identify new driving forces for economic development. China’s “demographic dividend”—the high labor force participation rate resulting from a proportionally high working population—is coming to an end, a shift that is accompanied by booming costs in terms of labor and land [6,7]. Gradual improvements in environmental regulations in China are also reducing the cost advantages it previously enjoyed in the labor-intensive industries, and Southeast Asian countries offer increased competition. Under these circumstances, China could easily fall into a middle-income trap, unless the transformation from resource-intensive to technology-intensive development [8], as well as a shift from dependence on factor inputs to innovation in factor productivity, is realized.

Rapid economic development demands intensive resource inputs. Under the hypothesis of “Environmental Kuznets Curve”, the massive energy consumption and CO_2_ emissions required to drive global economic growth have brought about irreversible climate change and environmental pollution [9], internationally. These environmental pollution and greenhouse gases that lead to environmental deterioration are regarded as undesirable outputs in economic activities. The unexpected output is inconsistent with the purpose of economic activities, which damages the social benefits. According to the World Energy Statistics Yearbook, which is released by BP [10], CO_2_ emission growth rates have hit record highs due to global energy consumption. Current progress in the transformation of the world energy system appears insufficient in meeting environmental demands and energy transformation goals. China’s economic development has been highly dependent on energy [11], and the country’s per capita energy use is much higher than the world consumption level. CO_2_ emissions in China were 9.429 billion tons in 2018, an increase of 2.053 billion tons compared with that in 2008, accounting for 27.8% of global CO_2_ emissions [12]. As the largest emitter, China is facing pressure concerning both emission abatement and environmental protection [13]. At present, the share of renewable energy is comparatively low in energy utilization structure, which greatly limits the level of innovation and sustainable development within China’s manufacturing industry. Facing the dual dilemmas of an economic development bottleneck and resource constraints, China urgently needs to identify effective technical approaches to the promotion of green innovation to realize sustainability.

How do the enterprises’ economic and innovation activities affect the green innovation efficiency and sustainable economy? Is the high-quality development transformation of the urban agglomeration region paying off? Based on these urgent to-be-solved problems, one of the most important purpose of our study is to scientifically assess the green innovation efficiency and the quantitative identification of its socioeconomic factors behind in the Pearl River Delta (PRD) urban agglomeration, we constructed an evaluation index system. Referring to existing literature, the concept of “green innovation efficiency” is generally recognized as a sophisticated innovation efficiency which can promote the coordinated growth of “economic–ecological–social system” [14,15,16,17]. Green innovation efficiency is always calculated as a ratio considering economic and environmental input and output of ogranization’s innovation activities [18]. It is a low-carbon index of innovation and environmental pollution that indicates the contribution of unit green innovation input to the output. Using a super slacks-based measure (Super-SBM) model, we evaluated the green innovation efficiency of the PRD cities, compared the efficiency of each city horizontally, and analyzed the change trend vertically within the study period. The socioeconomic factors of green innovation efficiency in the PRD urban agglomeration are analyzed by Tobit panel regression model. We ultimately putting forward a series of policy suggestions to help achieve the goals of industrial transformation and an improved economic development quality.

The remainder of the paper is organized in below: Section 2 discusses the related studies and existing literatures. The research methods together with the variables, data sources are presented in Section 3. Section 4 displays the empirical results of green innovation efficiency and the influential factors. Conclusions and policy suggestions are included in Section 5. All abbreviations are listed in Table A1.

## 2. Literature Review

Existing research into the innovation efficiency of the industrial sector and disparities in carbon efficiency between different geographical regions in China has reached a relatively mature stage. A general consensus exists amongst the majority of scholars regarding the existence of a notable regional unbalance in energy efficiency and carbon emission performance in China [19,20]. After evaluating the provincial-level carbon emission performance in China, Lin and Du [21] demonstrated that the eastern region performs better than the central or western regions, with their stochastic frontier model revealing obvious regional heterogeneity. Dong et al. [22] developed a comprehensive system for assessing regional carbon emission efficiency and proved that the eastern region ranks highest and the Western lowest among China’s three major geographical regions. Xiao et al. [23] investigated carbon emission performance in the Yangtze River Delta, revealing that upgrading the industrial structure is more carbon-friendly to the environment, especially the development of the tertiary industry. Shen et al. [24] identified a number of unbalanced characteristics in industrial carbon efficiency, using a low-carbon transformation index from the perspective of regional and industrial disparity. Despite the clear value of this previous work, study on the green innovation efficiency in relation to urban agglomerations remains underexplored. At the regional and national level, the PRD urban agglomeration has historically been characterized by labor-intensive and capital-intensive manufacturing [25]. As global competition has intensified, industrial clusters have gradually been integrated into the internal function of the region’s cities, and the urban agglomeration has been considered as the basic unit of participation in economic globalization.

Scholars analyzing the influencing factors driving improvements in industrial enterprise innovation efficiency—and the regional heterogeneity at work in those effects—have tended to conclude that environmental regulation is one of the pivotal influencing factors on green innovation efficiency. Strict environmental regulations limit the investment decisions and strategic choices of enterprises, resulting in increased production costs and investment risks [26]. Ultimately forcing enterprises to improve their market competitiveness through technological innovation [27,28]. Environmental regulation not only profits the development of ecological and social systems, but it can also place pressure on enterprises to transform and upgrade in order to ameliorate the additional expenditure brought about by the environmental regulation. Kneller and Manderson [29] proved that the pressure of environmental regulation pushes British manufacturing enterprises to expand the investment in environmental technology; however, the scholars noted, because such enterprises squeeze other R&D investment funds to do this, this behavior does not increase the total amount of R&D investment and promote capital accumulation. In comparison to this point of view, neoclassical economists hold a conservative attitude towards environmental regulation. They advocate that environmental regulation could play a promotive role on green innovation efficiency, innovation behaviors associated with regulation can offset additional costs, and enterprises can as a result even profit from environmental regulation [30,31]. Therefore, the combination of restrictive rules and incentive rules, coupled with green government procurement and ecological labeling system, can effectively promote green innovation behavior and the efficiency of enterprises [32]. Hashimoto and Haneda [33] found that the government’s environmental incentive policies can encourage enterprises’ green innovation behavior. Ming et al. [34] have similarly suggested that government technological expenditure and environmental regulation have been beneficial to the improvement of efficiency in the manufacturing sector.

In addition to the above focus on environmental regulation, a number of scholars have also addressed the influencing factors driving green innovation efficiency from micro and macro perspectives. From the micro perspective, factors such as enterprise scale, the effects of technology introduction, capital density, the industry profit rate, the environmental protection awareness of senior executives, consumer orientation, etc., have all be considered [23,35,36]. From the micro perspective, factors such as urban scale, economic openness, R&D input intensity, FDI, the industrial structure, energy intensity, government quality, etc., have also been examined [37].

In terms of the above summary of the related studies, the contributions of this paper lie in the following aspects: firstly, our research enriched and broadened the study in relation to green innovation efficiency in the industrial sector. To date, relevant works mainly concentrated on the regional and provincial level, while our studies addressing urban agglomerations and the city level. Secondly, the standard index evaluation system is at present difficult to quantify and unify, and it is difficult to determine the correct stage at which a given factor promotes or inhibits innovation efficiency. Thus, our paper comprehensive consideration of energy, environment, economy, innovation, and other factors; through this study, we build an index evaluation system for estimating green innovation efficiency in urban agglomerations. Thirdly, our findings complement existing research results gained considering the PRD, and scientifically represent the green innovation abilities of regional cities.

## 3. Methodology and Data

### 3.1. The Super-SBM Model

The frontier analysis method is widely used by researchers to measure technological efficiency [38]. Frontier analysis consists of two basic analysis methods: stochastic frontier analysis (SFA) and data envelopment analysis (DEA) [39]. In the process of evaluating innovation efficiency, model selection and improvement influence both data acquisition and result requirements, and also the capacity to modify the model by introducing other parameters. For measuring green innovation efficiency, two approaches can be adopted to construct an evaluation system. The first is to set criteria and dimensions in relation to green innovation efficiency, select an indicator system, and use the entropy method for giving the weight to variables [40]; the other is to use an SFA method based on an input-output perspective [41]. Applying these methods, a number of scholars have explored the relationship between technology and environmental efficiency [42]. Many earlier findings also emphasize the importance of policy limitations with regard to CO_2_ emission reduction [43,44].

Proposed by Charnes et al. [45], the DEA method is applied for compare the performance of multiple decision making units (DMUs). In its practical application, the DEA method has an absolute advantage both in the way that in deals with the efficiency evaluation problem in multi-input and multi-output analyses, and in the way that it effectively avoids issues arising from dimensionless data. For these reasons, scholars tend to choose the DEA method when solving efficiency issues related to technology [46,47]. The Super-SBM model, improved on the basis of the DEA model, is used to measure the green innovation efficiency of cities in the PRD. The fact that green innovation efficiency at the urban scale is affected by many geographical environment factors, rather than a single aspect of input or output, could potentially impact the evaluation results. The Super-SBM model, nevertheless, solves the problem of undesirable outputs. First proposed and improved by Tone [48], the model combines the merits of the DEA and SBM method, allowing researchers to add in undesirable outputs.

We assumed that the green innovation efficiency system is made up of *n* DMUs, and every unit contains inputs, desirable outputs, and undesirable outputs vectors [49]. Through *m* input factors, *s*_1_ desirable outputs and *s*_2_ undesirable outputs of every unit are produced. These input–output factors can be expressed as follows: $x\in R^{m},\;y^{d}\in R^{r_{1}},\;y^{u}\in R^{r_{2}}$. The matrix *X*, $y^{d}$, $y^{u}$ can be defined as:(1)$$X=\left[x_{1},\;\ldots ,\;x_{n}\right]\in R^{m\times n},\;y^{d}=\left[y_{1}^{d},\;\ldots ,\;y_{n}^{d}\right]\in R^{r_{1}\times n},\;\mathrm{and}\;y^{u}=\left[y_{1}^{u},\;\ldots ,\;y_{n}^{u}\right]\in R^{r_{2}\times n}.$$

Assuming these data are all positive, the SBM model can be written as follows [48]:$$\rho^{*}=min\frac{1-\frac{1}{m}\sum_{i=1}^{m}\frac{S_{i}^{-}}{X_{ik}}}{1+\frac{1}{r_{1}+r_{2}}\left(\sum_{p=1}^{r_{1}}\frac{S_{p}^{d}}{y_{pk}^{d}}+\sum_{q=1}^{r_{2}}\frac{S_{q}^{u}}{y_{qk}^{u}}\right)}.$$
(2)$$s.t\{\begin{array}{c} X_{ik}=\overset{n}{\underset{j=1}{\sum }}x_{ij}\lambda_{j}+S_{i}^{-}\\ y_{pk}^{d}=\overset{n}{\underset{j=1}{\sum }}y_{pj}^{d}\lambda_{j}-S_{p}^{d}\\ y_{qk}^{u}=\overset{n}{\underset{j=1}{\sum }}y_{qj}^{u}\lambda_{j}+S_{q}^{u}\\ \lambda_{j}>0,\;S_{i}^{-}\geq 0,S_{p}^{d}\geq 0,S_{q}^{u}\geq 0 \end{array}$$
where $X_{ik}$ denotes the *ith* input of ${\mathrm{DMU}}_{k}$, $Y_{pk}^{d}$ is desirable output, and $Y_{pk}^{u}$ presents undesirable output; *λ* is the weighted vector. The value of $\rho^{*}$ falls between 0 and 1. Only when $\rho^{*}=1$ and the slack variable meet the conditions when $S_{i}^{-}=0,\;S_{p}^{d}=0,\;S_{q}^{u}=0$, can the ${\mathrm{DMU}}_{k}\;\left(X_{ik},\;Y_{pk}^{d},\;Y_{pk}^{u}\right)$ be determined to be effective in the SBM model. While the SBM model cannot further rank the DMUs when their value are greater than 1, Tone [50] proposed the Super-SBM model. The corresponding Super-SBM model with undesirable outputs can be written as follows:$$\varphi^{*}=min\frac{\frac{1}{m}\sum_{i=1}^{m}\frac{\bar{x}}{X_{ik}}}{\frac{1}{r_{1}+r_{2}}\left(\sum_{p=1}^{r_{1}}\frac{{\bar{y}}^{d}}{y_{pk}^{d}}+\sum_{q=1}^{r_{2}}\frac{{\bar{y}}^{u}}{y_{qk}^{u}}\right)}$$
(3)$$s.t\{\begin{array}{c} \bar{x}\geq \overset{n}{\underset{j=1,\neq k}{\sum }}x_{ij}\lambda_{j}\\ {\bar{y}}^{d}\leq \overset{n}{\underset{j=1,\neq k}{\sum }}y_{pj}^{d}\lambda_{j}\\ {\bar{y}}^{u}\geq \overset{n}{\underset{j=1,\neq k}{\sum }}y_{qj}^{u}\lambda_{j}\\ \lambda_{j}>0,\;\bar{x}\geq x_{k},\;{\bar{y}}^{d}\leq y_{k}^{d},\;{\bar{y}}^{u}\geq y_{k}^{u} \end{array}$$

The objective function value of *φ** stands the efficiency of the DMU, which can be greater than 1. Other variables are defined similar to Equation (2). The results gained by applying this model to calculate the green innovation efficiency of the PRD cities align better with the actual statistics and provide a more comprehensive method to compare the DMUs performance.

Green innovation efficiency is a research object which takes the form of a comprehensive system. This paper selects indicators that can be categorized in terms of three distinct aspects: (1) input indicators, which should not only consider the three basic elements of labor, capital, and energy, but also the element of green energy; (2) desirable output indicators, which include research and development achievements and the economic benefits they bring; (3) undesirable output indicators, comprising of environmental pollution. In the literature on calculating production efficiency, the most important input factors are capital, labor, and energy. Referring to this existing research, we selected energy consumption per 10,000 yuan of GDP, R&D personnel, and capital stock of R&D internal expenditure, in order to represent the energy, labor, and capital input indicators, respectively. As for the desirable outputs, we selected two indicators—namely, the number of patent applications and the sales revenue of products of enterprises above a designated size—in order to represent scientific research level and achievement transformation ability. CO_2_ emissions, industrial wastewater emissions, and exhaust emissions are regarded as the undesirable output of economic activities. The resulting evaluation system of green innovation efficiency is presented in Table 1.

### 3.2. The Green Innovation Efficiency Evaluation Index System

To compare the undesirable outputs within the PRD cites, we employed a comprehensive index in order to measure pollutant emissions. Referring to previous research, a pollution emission index system for assessing pollutant production was therefore constructed. The weight of the three environmental pollutant indicators within the index was determined using an analytic hierarchy process (AHP), which examines and resolves synthesis-resolution issues [51]. Table 2. Index system for pollutant emissions. presenting the weights of industrial wastewater, industrial exhaust emissions, and CO_2_ emissions, which were found to be 0.6370, 0.2583, and 0.1047, respectively.

### 3.3. The Global Malmquist–Luenberger Index

The value of Malmquist index equal to the product of technology progress index and efficiency improvement index, first suggested by Caves et al. [52], which including technology progress and efficiency improvement. It has achieved connection between total factor production research and technical efficiency research simultaneously. Nevertheless, the Malmquist index cannot calculate total factor production in the presence of undesirable output, such as CO_2_ emission and air pollution. Chung et al. [53] proposed the Malmquist–Luenberger (ML) index on the foundation of directional distance function to solve this problem. Directional distance function is an evaluation method of estimating the relative efficiency of DMU along the predetermined direction vector without radial restriction. The ML index is widely used to measure the efficiency including the undesirable output as the advantage of need not to set the form of production function and the information of input–output cost, but only needs the number of input–output bundles and can be further decomposed into technological progress and efficiency improvement [54,55].

Firstly proposed by Malmquist [56], the Malmquist productivity index is applied to analyze the inputs consumption [49]. Oh [57] further proposed Global Malmquist–Luenberger (GML) index that on the basis of the ML index, and the GML index is written as follows:(4)$$GML_{t}^{t+1}=\left[\frac{1+D_{i}^{G}\left(x^{t},y^{t},b^{t}\right)}{1+D_{i}^{t}\left(x^{t},y^{t},b^{t}\right)}\times \frac{1+D_{i}^{t+1}(x^{t+1},y^{t+1},b^{t+1}}{1+D_{i}^{G}\left(x^{t+1},y^{t+1},b^{t+1}\right)}\right]\times \frac{1+D_{i}^{t}\left(x^{t},y^{t},b^{t}\right)}{1+D_{i}^{t+1}\left(x^{t+1},y^{t+1},b^{t+1}\right)}=GMLTECH_{t}^{t+1}\times GMLEFFCH_{t}^{t+1}.$$


In Equation (4), $D_{i}^{t}\left(x^{t},\;y^{t},\;b^{t}\right)$ and $D_{i}^{t+1}\left(x^{t+1},\;y^{t+1},\;b^{t+1}\right)\;$ presents the distance function of DMUs under the *t*, *t* + 1 time period, respectively. When $GML^{t,\;t+1}>1$, a productive capacity enables more desirable outputs and less undesirable outputs, implying productivity raise [57], and vice versa. The GML productivity index can be divided into technical change (TECH) and efficiency change (EFFCH). TECH reflects the change of technical efficiency by comparing the distance between the DUMs and the production frontier in different periods. In other words, the distance between the DUMs and the production frontier in different periods reflects the change of technical efficiency. The EFFCH is the ratio of the most productive level of the same input in different periods. Moreover, EFFCH can be further decomposed into pure technical efficiency and scale efficiency. Where $GMLTECH_{t}^{t+1}$ is the change of TECH of DMUs from period *t* to *t* + 1, $GMLEFFCH_{t}^{t+1}$ presents the variation of EFFCH from period *t* to *t* + 1. When $GMLTECH_{t}^{t+1}>1,\;GMLEFFCH_{t}^{t+1}>1$, the technology progress and the improvement of technology efficiency would promote the green innovation efficiency.

### 3.4. The Tobit Panel Regression Model

As defined above, the efficiency values estimated by the Super-SBM model are always greater than 0, which makes green innovation efficiency a limited dependent variable [58]. Given that the efficiency values of limited variables cannot be negative, a Tobit regression model was considered suitable in performing regression analysis on the influencing factors examined in this paper [59]. The Tobit model is widely used when there are many restrictive conditions for the type and data quality of dependent variables. However, the Tobit model also has its own insurmountable defects, it requires that the explanatory variables in the two-part model are not exactly the same [60]. In addition, the assumption in the system model is that random variables obey the joint normal distribution, and violating these two basic assumptions may lead to the model inestimability [61]. Therefore, it is very critical to scientifically set, estimate, and test Tobit model according to the research purpose and specific data. By maximum likelihood estimation [62], the basic Tobit model is constructed as follows:$$y^{*}=\beta X_{i}+\mu i$$
(5)$$y_{i}=\{\begin{array}{c} y_{i}^{*}\;\;if\;\;y_{i}^{*}>0\\ 0\;\;\;if\;\;y_{i}^{*}\leq 0 \end{array}\;$$where $y^{*}$ refers to the latent variable, $y_{i}$ is the limited dependent variable; $X_{i}$ stands the explanatory variable, β is the correlation coefficient; and μ presents the random error, i presents the ith DMU.

Based on the Equation (5), the regression model assumed that:(6)$$GIE_{i,t}=\beta_{0}+\beta_{1}IND_{i,t}+\beta_{2}FDI_{i,t}+\beta_{3}SCI_{i,t}+\beta_{4}EDU_{i,t}+\beta_{5}MOBI_{i,t}+\beta_{6}WAY_{i,t}+\varepsilon_{i,t}$$where $GIE_{i,t}$ is the value of green innovation efficiency in t year of ${\mathrm{DMU}}_{i}$, $IND_{i,t}$, $FDI_{i,t}$, $SCI_{i,t}$, and $MOBI_{i,t}$ stands for the output ratio of secondary industry, the actual utilization of foreign capital, the government’s R&D expenditure support level, and the penetration rate of the mobile phone network, respectively. $EDU_{i,t}$ and $WAY_{i,t}$ are the control variables of governmental education expenditure and highway mileage per capita. $\varepsilon_{i,t}$ is the random error term, and $\beta_{j}\left(j=0,\ldots 6\right)\;$ the constant coefficient.

### 3.5. Variables, Data Sources, and the Study Area

#### 3.5.1. Explanatory Variables

In order to examine the influencing factors driving green innovation efficiency levels in the PRD urban agglomeration, this study selected four explanatory variables, which reflect previous research undertaken from the dual perspectives of regional economy and government participation.

(1) Industrial structure

The emission of pollutants is affected by the industrial structure [63], which in turn affects green innovation efficiency values. Because the ratio of the output value of second industry (IND) can practically reflect the development degree of a given city’s manufacturing industry, IND is considered an appropriate measure of the influence exerted by of the industrial structure on the green innovation efficiency.

(2) Economic openness

Foreign capital increases the green innovation efficiency of regions in developing countries by means of capital and technology spillover effects from developed countries [64]. In contrast, research into industrial transfers that is based on the “pollution haven hypothesis” [65] recognizes that, without strong environmental regulation, inflows of foreign capital can result in high energy consumption and pollution [66], thus damaging local resource endowment and ecologies. The actual utilization of foreign capital (defined as FDI) is selected in this study to show how economic openness affects green innovation efficiency.

(3) Government expenditure

Enterprises are the most dynamic subjects in green innovation activities. The government’s financial support and policies are important factors in ensuring the quantity and quality of green innovation [67]. In addition, government financial expenditure on energy conservation and environmental protection can reflect their strength and capacities for pollution control and ecological protection. As such, we selected the proportion of science and technology expenditure (defined as SCI) to measure the government’s green innovation support level. In order to understand whether the education level indirectly affects green innovation efficiency, this paper also considers the proportion of government education expenditure (defined as EDU) as the control variable.

(4) Urban informationization level

The degree of information exchange and acquisition between cities depends on urban informatization [68], and the quality of urban digital infrastructure affects the flow of information technology resources. In this paper, the penetration rate of mobile phones (defined as MOBI) is selected as the index to measure the urban informatization level. Meanwhile, because non-digital infrastructures are also an important support in the process of urban informatization [69], we selected highway mileage per capita (defined as WAY) as a control variable to study the influence of infrastructure improvements on green innovation efficiency.

#### 3.5.2. Data Source and Study Area

This paper evaluates green innovation efficiency and investigates the socioeconomic factors in the PRD at the city level from 2009 to 2017. The 15 prefecture level cities considered by the study were Guangzhou, Shenzhen, Foshan, Dongguan, Jiangmen, Shaoguan, Huizhou, Heyuan, Qingyuan, Shanwei, Yunfu, Yangjiang, Zhuhai, Zhaoqing, and Zhongshan. As one of the country’s largest urban agglomerations, the PRD has developed both an advanced manufacturing industry and a modern service industry [70]. It is also the most populous urban agglomeration in China and possesses a significant degree of economic power and strength. For these reasons, the possibility to develop an innovative mode of ecological civilization and a high level of environmental friendliness in the PRD is considered to be high. Resource allocation capacity and management organization both need, however, to be further optimized if the region is to update the industry and drive future development.

All the original data used in our study were arranged from the Guangdong Statistical Yearbook, Guangdong Science, and Technology Statistical Yearbook, as well as the Cities Statistical Yearbook and Government Statistical Bulletin. The basic descriptive statistics are displayed in Table 3. In order to free from the multicollinearity problem, we used standard deviation to standardize all the variables.

## 4. Results and Discussion

### 4.1. Green Innovation Efficiency

The pollutant emission index (PEI) values are presented in Table 4. The PEI of the 15 cities of the PRD generally showed clear increases over the study period, with the exception of Foshan, Shanwei, Heyuan, and Yunfu, which all decreased slightly. The decrease witnessed in the period 2009–2010 can be attributed to the achievement of the goals of the 11th Five-Year Plan of Guangdong Province, through which the government put forward a series binding policies relating to energy consumption and pollutant emissions [71]. Between 2009 and 2017, the average PEI value of Heyuan was found to be the lowest (0.0171), a result which is closely associated with the tourism orientation of the city and its underdeveloped industrial system. This is further confirmed by the fact that the average PEI values of Guangzhou, Shenzhen, Foshan, and Dongguan—all cities with industrial clusters and highly developed economic levels—were much higher than other cities in the PRD. The correlation between economic development and environmental deterioration confirms the extensive economic development that has characterized the PRD in recent decades.

Applying a Super-SBM model with undesirable outputs, we measured the green innovation efficiency of 15 cities in the PRD, also analyzing the evolution characteristics of those levels over time. Table 5 shows results for the three years of 2009, 2013, and 2017, whereby GIE is green innovation efficiency, PTE represents pure technical efficiency, and SE is scale efficiency. The production frontier of each city’s annual data is generated by the static model, which is not the same, and the efficiency values of each year cannot be compared vertically. As such, we analyzed the cross-sectional data characteristics of each city’s green innovation efficiency.

Figure 1 displaying the change tendency of green innovation efficiency of 15 cities in the PRD. In 2009, obvious polarization was present in the green innovation efficiency levels of the region. Among the 9 cities characterized by an effective state of green innovation efficiency, Shenzhen was found to maintain significantly higher levels than other cities. The green innovation efficiency of the remaining eight cities was relatively balanced at a level less than 1, with the exception of Yangjiang. It is urgent for Yangjiang to increase investment in green innovation for further improving its efficiency. The six cities of Zhuhai, Shaoguan, Heyuan, Zhaoqing, Qingyuan, and Yunfu are shown to be characterized by an “inefficient” state of green innovation. With the exception of Zhuhai, these cities maintained green innovation efficiency levels of no more than 0.4. It means that the utilization and transformation level of low-carbon technology investment in these cities needs to be improved. The technical inefficiency of Zhuhai and Heyuan can be attributed to their low scale efficiency: Their lack of resource investment in innovation activities results in a situation of uneconomical scale. The technical inefficiency of Shaoguan and Yunfu, in contrast, is mainly due to their low level of pure technical efficiency, which in turn can be explained by a lack of maturity in both the allocation of innovation resources and activities by key economic actors. Zhaoqing and Qingyuan’s pure technical efficiency and scale efficiency were both found to be comparatively low—the pure technical efficiency of them were only 0.279 and 0.193, respectively. This suggests that the superior and local governments of the two cities need to attach more importance to various restrictive conditions and the construction of market-oriented systems. Meanwhile, enterprises in these cities also need to strengthen their organization and management capacities in order to achieve an optimal utilization of resources.

From 2010 to 2011, Zhuhai, Shaoguan, Zhaoqing, Qingyuan, and Yunfu were in the state of technical ineffectiveness among the PRD cities, which was less than that in 2009. While the polarization of green innovation efficiency is still obvious, among these five cities, the ineffective state of Zhuhai is due to the low scale efficiency. The others are mainly constrained by their obviously low pure technical efficiency, indicating that the utilization efficiency of innovative manpower and capital in these cities is in urgent need to improve. Meanwhile, through the value of cities’ scale efficiency, it reveals that government and enterprises still need to attach importance to the investment scale of green innovation resources in the PRD. Significantly, Foshan’s ineffective status changed that from pure technical efficiency to scale efficiency in 2014–2015, indicating that the city’s resource management and allocation level has been notably improved, as the investment in green innovation resources promotes the rapid development of green innovation growth.

In 2017, the number of cities characterized by ineffective levels of green innovation increased to eight. Huizhou entered this category for the first time in 2017, primarily as the result of limitations in the city’s pure technical efficiency. In addition, Zhuhai once again returned to an invalid state, reflecting its low scale efficiency. The green innovation efficiency value of Yangjiang again reached effective levels for three years in a row. We also note that the resource management level of Yunfu improved, although it was not matched by resources inputs commensurate with stimulating green innovation activities. The study also identified limitations in the growth of the green innovation efficiency of cities such as Heyuan and Zhuhai, which occurred because of a low scale efficiency from 2009 onwards.

### 4.2. The Global Malmquist-Luenberger Index Analysis

As the productivity index, the GML index is employed to evaluate dynamic variations in the green innovation efficiency of cities within the PRD. This allowed us to consider technological progress, and the measurement results of the GML index were set out in Table A2, where GML is the dynamic changes of green innovation efficiency, EC refers to the green innovation technical efficiency change, and TC stands the of green innovation technical change (the default base period efficiency is 1).

When undesirable outputs were considered within the indicator system, the green innovation efficiency of the PRD cities was shown to first decrease and then increase across the research period. The cumulative change value of the GML index for the 15 cities during the study period was found to be 1.302, indicating that the green innovation efficiency of the PRD cities increased by 30.2% over a period of 9 years (this result is shown in Figure 2. We further decomposed the GML index from the two perspectives of technological efficiency and technological progress. The cumulative change value of the technical efficiency index for the PRD urban agglomeration was 1.444, indicating an overall cumulative increase of 44.4% over the study period; and the cumulative change value of the technological progress index was 0.993, indicating an overall cumulative decrease of 0.7%. The empirical results suggest that the improvement of green innovation efficiency in the PRD mainly stem from improvements in technological efficiency, while the contribution of technological progress to overall efficiency was insignificant.

The results of our calculation of the GML and its decomposition index show that 9 cities—namely, Guangzhou, Shenzhen, Shaoguan, Heyuan, Dongguan, Zhongshan, Zhaoqing, Qingyuan, and Yunfu—were characterized by growth in their green innovation efficiency values during 2009–2017. The cumulative growth values of the GML index for Guangzhou, Heyuan, Qingyuan, and Yunfu were highest, reaching 1.849, 3.644, 3.033, and 2.232, respectively. The rapid growth of green innovation efficiency in Guangzhou is primarily due to technological progress, attesting to Guangzhou’s high level of R&D capacity and the great importance that is attached to high-tech development compared with other cities. To some extent, it should also be observed that Guangzhou neglected the utilization and management of existing green innovation technologies and available technical resources, which also directly hindered the further development of the city’s green innovation ability. Shenzhen is shown to be facing similar problems. The cumulative growth of green innovation efficiency in Heyuan, Qingyuan, and Yunfu was caused by increases in these cities’ technical efficiency. The development speed of technological R&D in these three cities remained, however, in the lower half of the 15 cities.

The green innovation of the PRD cities is still in the initial development stage. Compared with independent research and development, relevant enterprises and industries are more inclined to purchase and introduce mature and systematic technology from abroad. Besides, they prefer to invest capital, human, and other resources in the use of foreign technology, as well as strengthen environmental regulation, supervision and management. Thus, it results in the non-synchronization of technical utilization and technological progress [72]. To sum up, governments and enterprises of the PRD could transform less capital, labor, and energy into higher quality green innovation outputs by decreasing green innovation resources inputs and reducing the negative environmental effects in the process of green innovation activities. The decreases witnessed in the values of the technical progress index indicate that room for development exists in relation to the research and development abilities of cities in the PRD, in the process of pursuing green innovation behaviors through adjustments to inputs and outputs.

### 4.3. Panel Regression Results

Regression tests using the Tobit model were carried out after all variables were standardized; the results are shown in Table 6. All the influencing factors passed the significance test. Model 2 was achieved by adding in the control variables to Model 1. The results of the Tobit model are considered reasonable—key variables were found to maintain the same and significant influence on green innovation efficiency, with the exception of SCI. In Model 1, which is without control variables, IND, FDI, and SCI were significant at the level of 1% and MOBI was at the 5% significant level. After adding the control variables, all variables were significant at the level of 1%, except EDU, which was significant at the level of 5%.

The growth of the share of the secondary industry (IND) positively affects the promotion of green innovation efficiency. This is because, in line with the implementation of China’s strict policy of eliminating backward production facilities, the cities of the PRD have actively cultivated and developed new high-tech industries, which have delivered economic and environmental benefits. Within the industrial structure of the PRD, the electronics industry and light industry account for a large proportion of the region’s secondary industry. The light industry maintains the highest energy efficiency of all industries [73], and its limited energy consumption contributes greatly to the growth of the share of the secondary industry—as such, our results reveal that the internal structure of secondary industry in the PRD has gradually been optimized, as labor-intensive industries have been replaced by capital and technology-intensive ones. Hence, upgrading the industrial structure should be considered as an effective path to abate pollutant emissions.

The level of economic openness, which in this study was represented by FDI, was found to exert a prominent and positive influence on the increase of green innovation efficiency, indicating that the increased foreign capital investment can positively promote green innovation efficiency. This finding also reveals that the “Pollution Haven Hypothesis” is not supported in the PRD cities. On the one hand, this indicates that these cities rely on the introduction of green innovation technologies to achieve greater profits and meet ecological goals; on the other hand, this positive effect also benefits from recent policies and regulations that have sought to optimize the structure for handling introduced domestic and foreign capital in the region.

The proportion of government’s R&D expenditure (SCI) was shown to exert a negative effect on green innovation efficiency, whereby every 1% increase in the R&D expenditure is related to a 0.163% decrease in green innovation efficiency, in terms of the results of Model 2. We argue that the reasons for this can be understood from two distinct perspectives: namely those of local government and enterprises. Under the government system of fiscal decentralization, the assessment mechanism around GDP to some extent encourages local officials to ignore those side effects of economic development that are difficult to assess quantitatively. Meanwhile, due to its competitive advantage, government funds may squeeze out private funds and inhibit the green innovation vitality of small and medium-sized enterprises after entering the market. As for the enterprises, the essence of an enterprise is to maximize its economic benefits, which demands that enterprises try their best to reduce the cost of activities that cannot create economic performance. At present, the green environmental protection market is still in an initial phase in the PRD, and government investment in and support for green technological innovation is also very limited. On the other hand, some small and medium-sized enterprises have no clear understanding of their own development orientation and innovation ability, and lacking this planning, cannot effectively allocate and utilize resources.

The government’s education expenditure (EDU) is shown to negatively affect green innovation efficiency, a finding which demonstrates that the government’s investments in education are not being converted into technological achievements in time to improve the production efficiency. In addition, R&D institutions and universities in the PRD urban agglomeration should improve the transformation ability of their intellectual property. The urban informatization level, represented in this study by the variables of MOBI and WAY, was found to notably positive affects the improvement of green innovation efficiency. The construction of communication facilities and infrastructure is conducive to information and technology exchange between cities, thereby improving the green innovation efficiency of the region as a whole.

## 5. Conclusions

In the face of international and domestic pressure to address resource shortages and environment deterioration globally, green innovation represents an effective path for regional sustainable development. This paper calculated green innovation efficiency in the PRD urban agglomeration during 2009 to 2017, using a Super-SBM model, and examined the influencing factors which affect green innovation efficiency by way of a Tobit panel regression model. Our main conclusions are as below:

(1) By constructing a Super-SBM model, which was able to consider undesirable outputs, and analyzing the cross-sectional data, we found that the number of cities in the PRD which had achieved an “effective” state with respect to their green innovation efficiency levels decreased slightly during across the study period. In 2009, 9 of the 15 cities had reached an effective state; this number peaked in 2010, when a total of 10 cities had reached an effective state of green innovation efficiency. This basic situation was maintained until 2017, when it dropped to only 7–8 cities. In addition, polarization between cities in terms of their green innovation efficiency levels was also prominent, with the maximum value being more than 2, and the minimum value less than 0.1. These results show that the development of green innovation in the PRD urban agglomeration is significantly unbalanced, with the values of some cities fluctuating between effectiveness and invalidity, a sign that green innovation was to a certain extent being ignored in the process of urban development that characterized the region during this period, and that resources were in some cases being ineffectively allocated and managed, challenging the vitality of innovation activities;

(2) Decomposition analysis by the GML index, this study revealed that on the one hand, technological efficiency and technological progress have not kept pace with each other, which has restricted overall growth in green innovation efficiency. This means that the allocation and management of green innovation resources by the government gradually improved, and that to some extent the efficient utilization and intensive management of various resources was realized, to the extent that the scale of overall innovation investments became more economical and thus appropriate. On the other hand, most cities in the urban agglomeration were either characterized by negative green innovation efficiency growth, or by large fluctuations, demonstrating that it is hard to ensure the stable improvements of green innovation efficiency across a region;

(3) The regression results obtained using the Tobit panel model proved that the regional industrial structure, the extend of economic openness, and the city information level all exerted a significantly positive effect on the green innovation efficiency of the PRD cities. The level of government R&D and education expenditure were in contrast found to have negative impact on green innovation efficiency. The renewal of production technology and the introduction of capital and technology were found to play a vital role on promoting green innovation efficiency in the PRD urban agglomeration; governmental expenditure on scientific research, however, needs time to transform into productivity that promotes efficiency.

### Implications

Based on the results set out above, and the review of existing literature offered in opening, a series of meaningful and theoretical policy implications can be offered [58,74,75,76]. Our findings emphasize the importance of making clear the functional orientation and division of labor, a task that planning departments are well suited to. Against the background of unbalanced development in the PRD urban agglomeration, it is critical that cooperation and connection between cities and the exchange of technical knowledge are strengthened [77]. Management methods and resource allocation experience is required to establish durable bonds between cities. On the other hand, policy makers also need to focus on improving relevant policies for the flow of talent and funds between cities. The tremendous transregional flow of elements is the most obvious characteristic and advantage of an urban agglomeration. Based on their own functional orientation and division of labor, the well-developed central cities need to remove obstacles that hinder talent and capital flow. The underdeveloped edge cities need to attract high-tech talent to settle and increase employment rates through settlement policies, labor training, and social security. Given that regional disparities may impede such a catch-up process, uniformity in governmental regulatory regimes is also crucial [40]. Strengthening regional cooperation and connection is considered profitable to heighten green innovation efficiency.

To sum up, we suggest that policy makers guide government, enterprises, and scientific research institutions to invest in low-energy consumption and low-pollution technologies, through the formulation and improvement of legal systems. At the macro scale of the urban agglomeration itself, we should reasonably adjust the industrial location pattern and the industrial structure, so as to promote mutually beneficial ecological and economic conditions for enterprises and institutions, through the pursuit of innovative approaches. The establishment of special resource and environment groups and institutional departments could encourage the stricter enforcement of existing laws, and aid in the quantitative assessment of green innovation activities for relevant departments. Through transparent environmental information about matters such as pollutant discharge, water purification, and treatment, as well as programs for monitoring and disclosure, government departments, and enterprises can be encouraged to consciously abide by environmental laws and regulations. Besides, public conservation awareness and enthusiasm for participation in the implementation of policy can be strengthened.

There remains work to be done through future study of these issues. In the present paper, we measured green innovation efficiency and its socioeconomic factors of the PRD urban agglomeration; whether these effects were consistent in other urban agglomeration areas is unclear. Moreover, whether there are better indicators to measure green innovation efficiency is also well worth further research.

## Figures and Tables

**Figure 1 ijerph-18-12880-f001:**
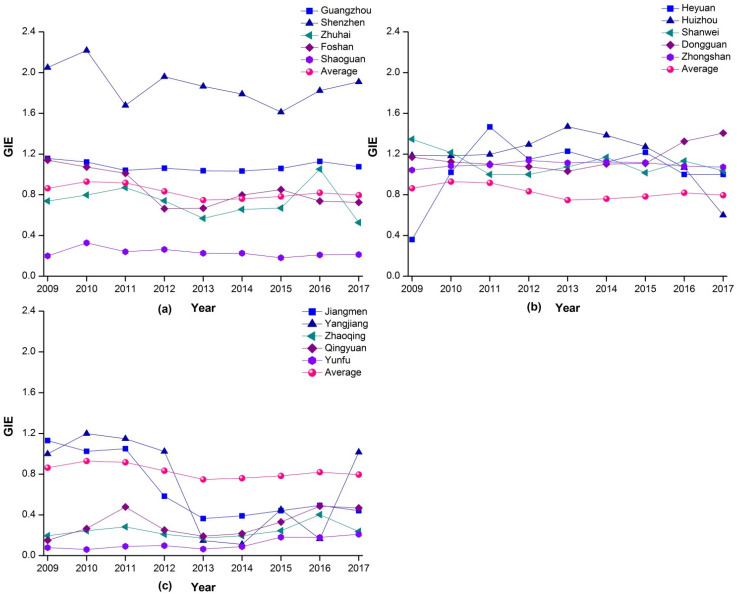
Change tendency of green innovation efficiency of the Pearl River Delta (PRD) cities, 2009–2017.

**Figure 2 ijerph-18-12880-f002:**
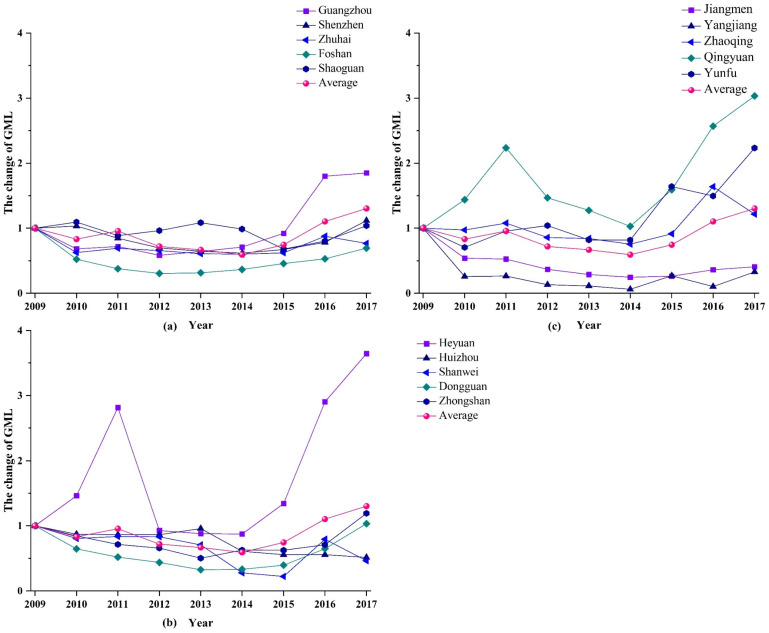
Cumulative change rates of Global Malmquist–Luenberger (GML) index in the Pearl River Delta urban agglomeration (2009–2017).

**Table 1 ijerph-18-12880-t001:** Green innovation efficiency (GIE) evaluation index system.

Variable Type	Evaluation Dimension	Indicators	Abbreviation
Inputs	Human resources investment	R&D personnel	HRI
	Capital investment	capital stock of R&D internal expenditure	CI
	Energy investment	energy consumption per 10,000 yuan of GDP	EI
Desirable output	Scientific research level	the number of patent applications	SRL
	Achievement transformation level	the sales revenue of new products of enterprises above designated size	ATL
Undesirable outputs	Pollutant emissions index	industrial wastewater emissions	PEI
		industrial exhaust emissions	
		CO_2_ emissions	

**Table 2 ijerph-18-12880-t002:** Index system for pollutant emissions.

Index Name	Indicators	Weight (%)
Pollutant emissions index	industrial wastewater emissions	63.70
	industrial exhaust emissions	25.83
	CO_2_ emissions	10.47

**Table 3 ijerph-18-12880-t003:** Descriptive statistics of indicators, 2009–2017.

Category	Variable	Units	Min.	Max.	Median	Mean	Std. Dev
Inputs	HRI	person	477	232,421	11,611	31,764.393	47,082.353
	CI	104 yuan	2595.705	29,546,545.750	424,515.061	2,287,651.372	4,718,560.295
	EI	ton/10^4^ yuan	0.363	1.737	0.582	0.683	0.280
Desirable outputs	SRL	pieces	172	177,103	5341	18,637.978	29,546.975
	ATL	10^4^ yuan	37,650.239	119,240,746.600	3,608,698.189	12,569,230.980	20,685,111.210
Undesirable outputs	PE	-	0.000	1.000	0.263	0.329	0.283
Influencing factors	IND	%	25.400	60.842	43.200	43.516	8.717
	FDI	10^4^ dollars	2808	740,129	80,394	146,075.974	179,241.920
	SCI	%	0.258	20.683	2.132	3.131	2.815
	MOBI	pieces/person	0.337	2.749	2.132	1.153	0.629
	EDU	%	1.415	28.396	20.911	20.337	4.951
	WAY	km/10^4^ person	1.304	64.939	24.389	25.930	18.586

**Table 4 ijerph-18-12880-t004:** Pollutant emission index values for 15 cities in the Pearl River Delta (2009–2017).

Cities	Pollutant Emission Index (PEI)								
	2009	2010	2011	2012	2013	2014	2015	2016	2017
Guangzhou	0.9077	0.8556	0.4432	0.8938	0.9218	0.8590	0.9536	1.0000	0.9920
Shenzhen	0.3567	0.2886	0.4143	0.5157	0.5833	0.5337	0.8164	0.5894	0.4822
Zhuhai	0.2289	0.1511	0.1811	0.1883	0.1926	0.1943	0.3274	0.2490	0.2599
Foshan	0.7179	0.6709	0.5945	0.6776	0.6346	0.5263	0.4248	0.6853	0.6073
Shaoguan	0.2925	0.2477	0.2139	0.3152	0.2633	0.2085	0.3106	0.4160	0.3528
Heyuan	0.0356	0.0160	0.0509	0.0183	0.0079	0.0092	0.0122	0.0022	0.0019
Huizhou	0.2234	0.1516	0.2789	0.2851	0.3061	0.2937	0.3679	0.3033	0.3731
Shanwei	0.0492	0.0213	0.0000	0.0051	0.0204	0.0141	0.0392	0.0134	0.0369
Dongguan	0.8974	0.8117	0.8716	0.9122	0.8975	0.9094	0.9209	0.8448	0.9493
Zhongshan	0.2762	0.2257	0.2416	0.2191	0.2498	0.2187	0.2586	0.3136	0.3030
Jiangmen	0.3259	0.2771	0.5312	0.4139	0.3862	0.4783	0.3577	0.3613	0.3879
Yangjiang	0.0035	0.0194	0.0394	0.0416	0.0561	0.0829	0.1239	0.0759	0.1212
Zhaoqing	0.1614	0.1613	0.2579	0.3375	0.3295	0.2980	0.3170	0.2995	0.2913
Qingyuan	0.1909	0.0960	0.2609	0.1536	0.1298	0.1712	0.1988	0.1399	0.1872
Yunfu	0.1512	0.0358	0.0629	0.0399	0.0416	0.0408	0.0234	0.0572	0.0628

**Table 5 ijerph-18-12880-t005:** Decomposition values for green innovation efficiency.

Cities	2009			2013			2017		
	GIE	PTE	SE	GIE	PTE	SE	GIE	PTE	SE
Guangzhou	1.157	1.168	0.990	1.036	1.047	0.989	1.076	1.100	0.979
Shenzhen	2.050	2.158	0.950	1.866	1.941	0.961	1.909	2.020	0.945
Zhuhai	0.738	1.027	0.718	0.568	1.021	0.556	0.527	1.028	0.513
Foshan	1.140	1.143	0.998	0.668	0.741	0.902	0.726	0.815	0.891
Shaoguan	0.199	0.223	0.893	0.226	0.306	0.740	0.213	0.272	0.782
Heyuan	0.360	1.165	0.309	1.228	3.074	0.399	1.000	1.000	1.000
Huizhou	1.187	1.213	0.978	1.469	1.502	0.978	0.601	0.720	0.834
Shanwei	1.346	1.601	0.841	1.074	1.422	0.755	1.035	1.222	0.847
Dongguan	1.170	1.172	0.999	1.032	1.047	0.985	1.406	1.414	0.994
Zhongshan	1.043	1.053	0.991	1.114	1.146	0.972	1.073	1.088	0.986
Jiangmen	1.131	1.133	0.999	0.364	0.479	0.76	0.441	0.627	0.702
Yangjiang	1.000	1.000	1.000	0.147	1.081	0.136	1.016	1.058	0.960
Zhaoqing	0.196	0.279	0.700	0.171	0.302	0.567	0.238	0.427	0.557
Qingyuan	0.150	0.193	0.777	0.191	0.306	0.622	0.468	0.528	0.887
Yunfu	0.079	0.091	0.864	0.064	0.101	0.639	0.209	1.028	0.203
Average	0.863	0.975	0.867	0.748	1.034	0.731	0.796	0.957	0.805

**Table 6 ijerph-18-12880-t006:** Tobit model regression results.

	Model 1				Model 2			
	Coefficient	Std. Error	t Value	Pr. (> |t|)	Coefficient	Std. Error	t Value	Pr. (> |t|)
IND	0.099 ***	0.035	2.85	0.005	0.180 ***	0.040	4.44	0.000
FDI	0.251 ***	0.061	4.12	0.000	0.298 ***	0.059	5.03	0.000
SCI	−0.120 ***	0.033	−3.62	0.000	−0.163 ***	0.036	−4.57	0.000
MOBI	0.167 **	0.065	2.58	0.011	0.282 ***	0.080	3.52	0.001
EDU					−0.086 **	0.043	−2.01	0.047
WAY					0.230 ***	0.069	3.33	0.001
Constant	0.792 ***	0.031	25.73	0.000	0.792 ***	0.029	27.12	0.000
R^2^	0.5795				0.6498			
N	117				117			

Notes: **, *** indicate 5% and 1% significance level, respectively.

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
