# Peer review of "Evaluating Green Innovation Efficiency and Its Socioeconomic Factors Using a Slack-Based Measure with Environmental Undesirable Outputs"

_ijerph, 2021, doi:10.3390/ijerph182412880_

Round 1
Reviewer 1 Report
Thank you for giving me the opportunity to review this interesting paper. The paper evaluates the green innovation efficiency of 15 cities in the Pearl River Delta using non-parametric techniques. The use of panel data allows the authors to expand their efficiency analysis into the measurement of productivity and its drivers. Finally, the authors use truncated regression to evaluate the impact on several socio-economic factors on green innovation efficiency. The paper is very well written, however, my recommendation is major revisions. My main concern is that the authors need to clarify the use of the models in their study. Moreover, they need to clarify the terminology of the GML productivity indicator. I believe that the GML measures green innovation productivity change.
I provide my comments below.
Introduction
- Could the authors please write a few sentences explaining why they performed their research on the PRD cities?
- Are any previous studies that measured green innovation efficiency in PDR or elsewhere?
- - Could the authors pelase explain a bit more the concept of green innovation efficiency?
- - The authors used a productivity indicator including several undesirable outputs such as CO2. However, this concept is not discussed in the introduction. Could the authors please explain why?
- - At the moment it is not clear to me the gaps to the existing literature that this study aims to fill. Could the authors please elaborate on this? Literature review It is well written and I don't have any further comments.
Methodology
- - Could the authors please explain the advanatges of using DEA instead of SFA or other techniques such as partial frontier ones?
- - Could the authors please explain why they used the super efficiency model? Why did not they use the traditional SBM model?
- - Directional distance functions take into account undesirable outputs. Could the authors please explain why they used SBM instead of directional distance functions?
- - Could the authors confirm if they ran the model under CRS or VRS? I am kindly asking this because it is not clear to me if they measured scale efficiency and scale efficiency change.
- - Could the authors please better explain the concepts of technical change and technical efficiency change in the GML productivity indicator?
- - Could the authors confirm if the GML measures green innovation productivity? I believe that the GML is a productivity indicator so the authors measure green innovation productivity change. The green innovation productivity indicator is further decomposed into green innovation efficiency change (GIEC) and green innovation technical change (GITC).
- - Could the authors please briefly explain the concept of directional distance functions? Which models were solved to calculate the components of the GML productivity indicator?
- - Could the authors please explain why they used the term "global"? DId they pool all the data together to deal with the small number of observations relative to the number of variables used in the study (i.e. to fulfil Cooper's rule)?
- - Simar and Wilson (2007) highlighted that Tobit regression is not appropriate for exploring the impact of operating characteristics on efficiency scores. This is because of serrial correlation between efficiency scores, explanatory variables and error. Therefore, the authors reccommended using bootstrap truncated regression instead of Tobit regression. Could the authors add the limitations of Tobit regression and briefly discuss potential alternatives? They don't have to re-run the models
- - just a discussion of the limitations of Tobit regression.
Results and discussion
- - Could the authors please explain how the reader should interpret an efficiency score of 2.050?
- - Does Fig.2 refer to GIE or green innovation productivity change?
- - Could the authors explain the high GIE of Heyuan from 2015 onwards, i.e. Fig 2b.
- - I am not convinced why the authors refer to GML as green innovation efficiency indicator and not a productivity indicator. I think that the label of the tables and figures is confusing.
- - Does Figure 2 report the results of GML or EC? If the results refer to EC, then it is ok because they measure the change in green innovation efficiency. If the results refer to GML, then the figures should not be labelled as change in GIE. They measured change in green innovation productivity.
- - Could the authors confirm if they measured green innovation scale efficiency change as well? If yes, why didn't they report this information?
Conclusions I don't have any further comments
Author Response
Response to Reviewer 1 Comments
Reviewer #1: Thank you for giving me the opportunity to review this interesting paper. The paper evaluates the green innovation efficiency of 15 cities in the Pearl River Delta using non-parametric techniques. The use of panel data allows the authors to expand their efficiency analysis into the measurement of productivity and its drivers. Finally, the authors use truncated regression to evaluate the impact on several socio-economic factors on green innovation efficiency. The paper is very well written, however, my recommendation is major revisions. My main concern is that the authors need to clarify the use of the models in their study. Moreover, they need to clarify the terminology of the GML productivity indicator. I believe that the GML measures green innovation productivity change.
I provide my comments below.
- Introduction
Point 1: Could the authors please write a few sentences explaining why they performed their research on the PRD cities?
Response 1:Thank you very much for your suggestions. we have already explained why we performed our research on the PRD cities. And we clearly stated in the part of “3.5.2 Data source and study area” as:
“This paper evaluates green innovation efficiency and investigates the socioeco-nomic factors in the PRD at the city level from 2009 to 2017. The 15 prefecture level cities considered by the study are Guangzhou, Shenzhen, Foshan, Dongguan, Jiangmen, Shaoguan, Huizhou, Heyuan, Qingyuan, Shanwei, Yunfu, Yangjiang, Zhuhai, Zhaoqing, and Zhongshan. As one of the country’s largest urban agglomerations, the PRD has developed both an advanced manufacturing industry and a modern service industry. It is also the most populous urban agglomeration in China and possesses a significant degree of economic power and strength. For these reasons, the possibility to develop an innovative mode of ecological civilization and a high level of environ-mental friendliness in the PRD is considered to be high. Resource allocation capacity and management organization both need, however, to be further optimized if the re-gion is to update the industry and drive future development.”.
Point 2: Are any previous studies that measured green innovation efficiency in PDR or elsewhere?
Response 2:Thank you very much for your suggestions. There are a lot of work have been down in the research field of green innovation efficiency. However, none of the papers involved taking the cities in the Pearl River Delta as the research region. And we list some relevant reference in below for your reference:
[1] Fan F., Lian H., Liu XY., Wang XL. Research on regional differences and influencing factors of green technology innovation efficiency of China's high-tech industry, Journal of Cleaner Production. 2021(287): 125060. DOI: 10.1016/j.jclepro.2020.125060
[2] Zhang JX., Kang L., Li H. et al., The impact of environmental regulations on urban Green innovation efficiency: The case of Xi'an, Sustainable Cities and Society. 2020(57): 102123. DOI: 10.1016/j.scs.2020.102123
[3] Research on regional differences and influencing factors of green technology innovation efficiency of China's high-tech industry, Journal of Computational and Applied Mathematics. 2020(369): 112597. DOI: 10.1016/j.cam.2019.112597
[4] Sun HP., Edziah BK., Kporsu AK. Institutional quality, green innovation and energy efficiency, Energy Policy. 2019(135): 111002. DOI10.1016/j.enpol.2019.111002
Point 3: Could the authors please explain a bit more the concept of?
Response 3:Thank you very much for your comments. According to your suggestions, we have given a more detailed explanation of the concept of green innovation efficiency in the part of Introduction, and the corresponding content are appended in below:
How do the enterprises’ economic and innovation activities affect the green innovation efficiency and sustainable economy? Is the “middle-income trap” exists in the region and cities with developed resource-intensive or labor-intensive industries? And whether the high-quality development transformation of the urban agglomeration region has been paying off? Based on these urgent to be solved problems, one of the most important purpose of our study is to scientifically assess the green innovation efficiency and the quantitative identification of its socioeconomic factors behind in the Pearl River Delta (PRD) urban agglomeration, we constructed an evaluation index system. Referring to existing literature, the concept of “green innovation efficiency” is generally recognized as a sophisticated innovation efficiency which can promote the coordinated growth of “economic-ecological-social system” [14-17]. Green innovation efficiency is always calculated as a ratio considering economic and environmental input and output of ogranization’s innovation activities[18]. It is a low-carbon index of innovation and environmental pollution that indicates the contribution of unit green innovation input to the output.
Point 4: The authors used a productivity indicator including several undesirable outputs such as CO2. However, this concept is not discussed in the introduction. Could the authors please explain why?
Response 4:Thank you very much for your suggestions. Undesirable outputs, also named as unexpected output, is an element often involved in economics and econometric models. We did ignore the introduction of unexpected output. According to your suggestions, we briefly introduced and discussed this concept in the corresponding part of the Introduction, and the revised part appended in below:
Rapid economic development demands intensive resource inputs. Under the hypothesis of “Environmental Kuznets Curve”, the massive energy consumption and CO2 emissions required to drive global economic growth have brought about irreversible climate change and environmental pollution, internationally. These environmental pollution and greenhouse gases that lead to environmental deterioration are regarded as undesirable outputs in economic activities. And the unexpected output is inconsistent with the purpose of economic activities, which damages the social benefits. According to the World Energy Statistics Yearbook, which is released by BP [9], CO2 emission growth rates have hit record highs due to global energy consumption. Current progress in the transformation of the world energy system appears insufficient in meeting environmental demands and energy transformation goals. China’s economic development has been highly dependent on energy, and the country’s per capita energy use is much higher than the world consumption level. CO2 emissions in China were 9.429 billion tons in 2018, an increase of 2.053 billion tons compared with that in 2008, accounting for 27.8% of global CO2 emissions [10]. As the largest emitter, China is facing pressure concerning both emission abatement and environmental protection [11]. At present, the share of renewable energy is comparatively low in energy utilization structure, which greatly limits the level of innovation and sustainable development within China’s manufacturing industry. Facing the dual dilemmas of an economic development bottleneck and resource constraints, China urgently needs to identify effective technical approaches to the promotion of green innovation to realize sustainability.
Point 5: At the moment it is not clear to me the gaps to the existing literature that this study aims to fill. Could the authors please elaborate on this? Literature review It is well written and I don't have any further comments.
Response 5:Thank you very much for your suggestions. In the last part of the Literature review, we have explained the contribution of our work. In order to make readers more clearly to readers, we elaborate and emphasize the innovation and contribution of this paper in detail. And the corresponding manuscript is modified as follows:
In terms of the above summary of the related studies, the contributions of this paper lie in the following aspects: firstly, our research enriched and broadened the study in relation to green innovation efficiency in the industrial sector. To date relevant works mainly concentrated on the regional and provincial level, while our studies addressing urban agglomerations and the city level. Secondly, the standard index evaluation system is at present difficult to quantify and unify, and it is difficult to determine the correct stage at which a given factor promotes or inhibits innovation efficiency. Thus, our paper comprehensive consideration of energy, environment, economy, innovation, and other factors; through this study, we build an index evaluation system for estimating green innovation efficiency in urban agglomerations. Thirdly, our findings complement existing research results gained considering the PRD, and scientifically represent the green innovation abilities of regional cities.
- Methodology
Point 6: Could the authors please explain the advantages of using DEA instead of SFA or other techniques such as partial frontier ones?
Response 6:Thank you very much for your suggestions. SFA and DEA both have their own advantages and limitations. We cannot simply conclude that one is better than the other. Therefore, we must make judgments according to specific problems and actual measurement results. Firstly, compared with DEA model, the basic assumptions of SFA model are more complex, and the specific forms of production function and technical inefficiency term distribution need to be considered, which directly leads to the difficulty of further expansion of the model. The main advantage of DEA method is that it does not need to consider the specific form of production frontier, only needs input-output data, and the model is easy to be extended in other forms. Secondly, one of the advantages of DEA is that it can directly deal with multiple outputs, while SFA is more complex. As the SFA model needs to combine multiple outputs into a comprehensive output, or solve it by using distance function. Moreover, because the parameters estimated by the maximum likelihood estimation method used in SFA model have the consistency of large samples, SFA is more suitable for large sample calculation. While our research based on a small sample data, DEA method is more scientific and efficient in our study. In summary, this paper chose the SBM model based on the principles of DEA method to estimate the efficiency.
Point 7: Could the authors please explain why they used the super efficiency model? Why did not they use the traditional SBM model?
Response 7:Thank you very much for your comments. The reasons why we used the super-SBM method instead of traditional SBM model have been described in detail in the part of “3. Methodology and data” as “The Super-SBM model, improved on the basis of the DEA model, is used to measure the green innovation efficiency of cities in the PRD. The fact that green innovation efficiency at the urban scale is affected by many geographical environment factors, rather than a single aspect of input or output, could potentially impact the evaluation results. The Super-SBM model, nevertheless, solves the problem of undesirable outputs.” and “While the SBM model can’t further rank the DMUs when their value are greater than 1, Tone [50] proposed the Super-SBM model.”.
Point 8: Directional distance functions take into account undesirable outputs. Could the authors please explain why they used SBM instead of directional distance functions?
Response 8:Thank you very much for your comments. The directional distance function model is a generalization of the radial model. It can easily deal with the situation of undesirable outputs, but its efficiency measurement does not solve the problem of unit invariance, which is an obstacle to the application of directional distance function in practice. The data standardization principle of SBM model provides a general method to maintain the unit invariance for the efficiency measurement method. At the same time, after using the standardized data, the efficiency measurement results of radial and non-radial models remain unchanged. Based on this, we intend to use the SBM model instead of directional distance functions.
Point 9: Could the authors confirm if they ran the model under CRS or VRS? I am kindly asking this because it is not clear to me if they measured scale efficiency and scale efficiency change.
Response 9:Thank you very much for your comments. We confirm that we run the BCC model under the assumption of VRS. And we have presented the results of pure technical efficiency and scale efficiency under VRS condition.
Point 10: Could the authors please better explain the concepts of technical change and technical efficiency change in the GML productivity indicator?
Response 10:Thank you very much for your comments. According to your suggestions, we added the concepts of technical change and technical efficiency change in the part of “3.3 The Global Malmquist-Luenberger index”. And the revised manuscript are as follows:
In Equation (4), and presents the distance function of DMUs under the t, t+1 time period respectively. When , a productive capacity enables more desirable outputs and less undesirable outputs, implying productivity raise [58], and vice versa. GML productivity index can be divided into technical change (TECH) and efficiency change (EFFCH). TECH reflects the change of technical efficiency by comparing the distance between the DUMs and the production frontier in different periods. In other words, the distance between the DUMs and the production frontier in different periods reflects the change of technical efficiency. And the EFFCH is the ratio of the most productive level of the same input in different periods. Moreover, EFFCH can be further decomposed into pure technical efficiency and scale efficiency. Where is the change of TECH of DMUs from period t to t+1, presents the variation of from period t to t+1. When, the technology progress and the improvement of technology efficiency would promote the green innovation efficiency.
Point 11: Could the authors confirm if the GML measures green innovation productivity? I believe that the GML is a productivity indicator so the authors measure green innovation productivity change. The green innovation productivity indicator is further decomposed into green innovation efficiency change (GIEC) and green innovation technical change (GITC).
Response 11:Thank you very much for your comments. We confirm that the GML index is used to measure the change of green innovation productivity. In the part of “4.2 The Global Malmquist-Luenberger index analysis”, we clearly explained that “The GML index is employed to evaluate dynamic variations in the green innovation efficiency of cities within the PRD.” And the green innovation productivity indicator in this paper has been further decomposed into green innovation efficiency change (abbreviated as EC) and green innovation technical change (abbreviated as TC).
Point 12: Could the authors please briefly explain the concept of directional distance functions? Which models were solved to calculate the components of the GML productivity indicator?
Response 12:Thanks very much for your suggestions. Directional distance function is an evaluation method of estimating the relative efficiency of decision-making unit (DMU) along the predetermined direction vector without radial restriction. The specific calculation principle was written in Equation (4). According to your suggestions, we briefly explained this in the paper, and the relevant modifications are as follows:
The value of Malmquist index equal to the product of technology progress index and efficiency improvement index, first suggested by Caves et al. [52], which including technology progress and efficiency improvement. It has achieved connection between total factor production research and technical efficiency research simultaneously. Nevertheless, Malmquist index cannot calculate total factor production in the presence of undesirable output, such as CO2 emission and air pollution. Chung et al. [53] proposed the Malmquist-Luenberger (ML) index on the foundation of Directional Distance Function to solve this problem. Directional distance function is an evaluation method of estimating the relative efficiency of DMU along the predetermined direction vector without radial restriction. The ML index are widely used to measure the efficiency including the undesirable output as the advantage of needn’t to set the form of production function and the information of input-output cost, but only needs the number of input-output bundles and can be further decomposed into technological progress and efficiency improvement [54,55].
Point 13: Could the authors please explain why they used the term "global"? Did they pool all the data together to deal with the small number of observations relative to the number of variables used in the study (i.e. to fulfil Cooper's rule)?
Response 13:Thank you very much for your comments. The reason why we used the Global Malmquist-Luenberger (GML) is that the essence of GML index lies in the construction of the global frontiers. That is, the production possibility set of each stage can form a global frontier by envelope method. Thus, the global directional distance function can be obtained. The traditional ML index calculation uses the directional distance function, but there are some problems in linear programming, such as no solution, not meeting the conditions of transitivity and additivity. GML index method can be used to overcome the above defects, that why we used the term “global”. And we have pooled all the data together to deal with the small number of observations relative to the number of variables used in this paper.
Point 14: Simar and Wilson (2007) highlighted that Tobit regression is not appropriate for exploring the impact of operating characteristics on efficiency scores. This is because of serial correlation between efficiency scores, explanatory variables and error. Therefore, the authors recommended using bootstrap truncated regression instead of Tobit regression. Could the authors add the limitations of Tobit regression and briefly discuss potential alternatives? They don't have to re-run the models, just a discussion of the limitations of Tobit regression.
Response 14:Thank you very much for your insights and suggestions on the model. According to your suggestions, we added briefly discussion about both the advantageous and limitations of Tobit regression in the corresponding part of our paper. And the revised contents are as follows:
As defined above, the efficiency values estimated by the Super-SBM model are always greater than 0, which makes green innovation efficiency a limited dependent variable [59]. Given that the efficiency values of limited variables cannot be negative, a Tobit regression model was considered suitable in performing regression analysis on the influencing factors examined in this paper [60]. Tobit model is widely used when there are many restrictive conditions for the type and data quality of dependent variables. However, Tobit model also has its own insurmountable defects, it requires that the explanatory variables in the Two-Part model are not exactly the same [61]. In addition, the assumption in system model of random variables obey the joint normal distribution, violating these two basic assumptions may lead to the model inestimability [62]. Therefore, it is very critical to scientifically set, estimate and test Tobit model according to the research purpose and specific data.
- Results and discussion
Point 15: Could the authors please explain how the reader should interpret an efficiency score of 2.050?
Response 15:Thank you very much for your comments. The 2.050 refers to the score of green innovation efficiency (GIE) estimated by the SBM model. When the score of GIE greater than 1, it means that the GIE of city has been enhanced. In our study, the GIE score of 2.050 indicates that the city is in a stable state of high green innovation level.
Point 16: Does Fig.2 refer to GIE or green innovation productivity change?
Response 16:Thank you very much for your comments. Fig. 2 refers to the cumulative change value of green innovation efficiency.
Point 17: Could the authors explain the high GIE of Heyuan from 2015 onwards, i.e. Fig 2b.
Response 17:Thank you very much for your comments. As we briefly explained that the high GIE of Heyuan as: “The cumulative growth of green innovation efficiency in Heyuan, Qingyuan, and Yunfu was caused by increases in these cities’ technical efficiency.” In fact, the GIE increase obviously in the period of 2015-2017. Since the 13th Five-Year Plan of Guangdong province, the policies of economic development focus on the new and high technology through vigorously exploration in the fields of the Internet plus, innovation system, low-carbon industries and so on. The reason behind the high GIE of Heyuan is that the number of green industry enterprises in Heyuan has increased notably, and the development of green industry has achieved remarkable fruits. The innovation activities, green and low pollution production of these enterprises have greatly promoted the improvement of GIE.
Point 18: I am not convinced why the authors refer to GML as green innovation efficiency indicator and not a productivity indicator. I think that the label of the tables and figures is confusing.
Response 18:Thanks much for your suggestions. In this study, the green innovation efficiency is considered as a technological gap ratio measured by the Super-SBM model, and we agree that GML is a productivity indicator which reflects the dynamic changes of the green innovation efficiency. In order to avoid misunderstanding of the table title to readers, we have modified the table name related to GML index in the article. Therefore, we changed the name of Table A2 to “Table A2. The GML, EC and TC index of cities in the PRD.”. In addition, we altered the label in the Fig.2. and modified the axis label of this graph.
Point 19: Does Figure 2 report the results of GML or EC? If the results refer to EC, then it is ok because they measure the change in green innovation efficiency. If the results refer to GML, then the figures should not be labelled as change in GIE. They measured change in green innovation productivity.
Response 19:Thank you very much for your comments. The Fig.2 report the results of GML index in PRD cities. According to your suggestions, we altered the label in the Fig.2. and modified the axis label of this graph appended in below:
Fig. 1. Cumulative change rates of GML index in the Pearl River Delta urban agglomeration (2009-2017)
Point 20: Could the authors confirm if they measured green innovation scale efficiency change as well? If yes, why didn't they report this information?
Response 20:Thank you very much for your comments. As the change of scale efficiency of green innovation is not the focus of our research, so the results are not reported.
4.Conclusions
Point 21: I don't have any further comments
Response 21: Thank you very much for your recognition and appreciation of our work.
Once again, we appreciate sincerely for the reviewers’ warm and patient work, and thank you very much for your constructive comments and suggestions which really promoted the quality of our manuscript. We hope that our earnest revision will conform to the approval of International Journal of Environmental Research and Public Health.
*********************************************************************
References
Berkel Rv. Eco-Innovation: opportunities for advancing waste prevention. International Journal of Environmental Technology and Management. 2007;7:527-50.
Chen YS, Lai SB, Wen CT. The Influence of Green Innovation Performance on Corporate Advantage in Taiwan. Journal of Business Ethics. 2006;67:p.331-9.
Oltra V, Jean MS. Sectoral Systems Of Environmental Innovation: An Application To The French Automotive Industry. Technological Forecasting & Social Change. 2009;76:567-83.
Publishing O. OECD Green Growth Studies Towards Green Growth: Monitoring Progress: OECD Indicators. Sourceoecd Environment & Sustainable Development. 2011:i-146.
Fan F, Lian H, Liu X, Wang X. Can environmental regulation promote urban green innovation Efficiency? An empirical study based on Chinese cities. Journal of Cleaner Production. 2021;287.
- Statistical Review of World Energy. In: p.l.c. B, editor. UK: BP; 2019. p. 1-68.
IEA. CO2 Emissions from Fuel Combustion 2019 Highlights. Paris: IEA; 2019.
Huang J, Dua D, Hao Y. The driving forces of the change in China's energy intensity: An empirical research using DEA-Malmquist and spatial panel estimations. Economic Modelling. 2017;65:41-50.
Tone K. A slacks-based measure of super-efficiency in data envelopment analysis. European Journal of Operational Research. 2002;143:32-41.
Oh DH. A global Malmquist-Luenberger productivity index. Journal of Productivity Analysis. 2010;34:183-97.
Caves DW, Christensen LR, Diewert WE. Multilateral Comparisons of Output, Input, and Productivity Using Superlative Index. The Economic Journal. 1982;92:73-86.
Chung Y, Fare R, Grosskopf S. Productivity and Undesirable Outputs: A Directional Distance Function Approach. Journal of Environmental Management. 1997;51:229-40.
Kortelainen M. Dynamic environmental performance analysis: A Malmquist index approach. Ecological Economics. 2008;64:p.701-15.
Zhou P, Ang BW, Han JY. Total factor carbon emission performance: A Malmquist index analysis. Energy Economics. 2010;32:194-201.
Shuai S, Fan Z. Modeling the role of environmental regulations in regional green economy efficiency of China: Empirical evidence from super efficiency DEA-Tobit model. J Environ Manage. 2020;261:110227.
Zeng L, Lu H, Liu Y, Zhou Y, Hu H. Analysis of Regional Differences and Influencing Factors on China’s Carbon Emission Efficiency in 2005–2015. Energies. 2019;12:3081-102.
Takeshi A. The Estimation of a Simultaneous-Equation Tobit Model. International Economic Review. 1979;20:169-81.
Ahn H, L.Powell J. Semiparametric estimation of censored selection models with a nonparametric selection mechanism. Journal of Econometrics. 1993;58:3-29.
Cuesta RA, Lovell CAK, Zofio JL. Environmental efficiency measurement with translog distance functions: A parametric approach. Ecological Economics. 2009;68:2232-42.
Hattori T. Relative Performance of U.S. and Japanese Electricity Distribution: An Application of Stochastic Frontier Analysis. Journal of Productivity Analysis. 2003;19:115-.
Huang J, Yu Y, Ma C. Energy Efficiency Convergence in China: Catch-Up, Lock-In and Regulatory Uniformity. Environ Resource Econ. 2018;70:107-30.
Lin B, Du K. Technology gap and China's regional energy efficiency: A parametric metafrontier approach. Energy Economics. 2013;40:529-36.
Zhou Z, Liu C, Zeng X, Jiang Y, Liu W. Carbon emission performance evaluation and allocation in Chinese cities. Journal of Cleaner Production. 2018;172:1254-72.
Cheng Z, Li W. Independent R and D, Technology Introduction, and Green Growth in China’s Manufacturing. Sustainability. 2018;10.

Reviewer 2 Report
The manuscript submitted for review was prepared with due care and based on a thorough knowledge of the subject. However, there are some comments I would like to make to the authors:
Abstract: prepared in accordance with current scientific standards.
Introduction section: The introduction should state the purpose of the work in the form of the research problem supported by a hypothesis or a set of questions, explaining briefly the methodological approach used to examine the research problem.
Literature review section The part from the lines: 118-145 should be shortened and move to the Methodology and data section.
Methodology and data section: a really good part of the paper.
Results and Discussion section: You should interpret and describe the significance of the findings in light of what was already known about the research problem being investigated and to explain any new understanding or insights that emerged as a result of your study of the problem. The discussion should always connect to the introduction by way of the research questions or hypotheses you posed.
Conclusions section: I have no comments on this part.
In conclusion, the article requires some revisions before being published.
Author Response
Response to Reviewer 2 Comments
Reviewer #2: The manuscript submitted for review was prepared with due care and based on a thorough knowledge of the subject. However, there are some comments I would like to make to the authors:
Point 1: Abstract: Prepared in accordance with current scientific standards.
Response 1: Thank you very much for your suggestions. According to your comments, we have carefully re-checked the expression and format of the abstract to ensure this part in accordance with current scientific standards of the IJERPH.
Point 2: Introduction section: The introduction should state the purpose of the work in the form of the research problem supported by a hypothesis or a set of questions, explaining briefly the methodological approach used to examine the research problem.
Response 2: Thank you very much for your suggestions. According your advice, we set several questions before we state our research purpose. Moreover, we have been briefly introduced the methodological approach we used to examine the research problem in the part of the Introduction. And the revised manuscript appended in below:
Rapid economic development demands intensive resource inputs. Under the hypothesis of “Environmental Kuznets Curve”, the massive energy consumption and CO2 emissions required to drive global economic growth have brought about irreversible climate change and environmental pollution, internationally. These environmental pollution and greenhouse gases that lead to environmental deterioration are regarded as undesirable outputs in economic activities. And the unexpected output is inconsistent with the purpose of economic activities, which damages the social benefits. According to the World Energy Statistics Yearbook, which is released by BP [9], CO2 emission growth rates have hit record highs due to global energy consumption. Current progress in the transformation of the world energy system appears insufficient in meeting environmental demands and energy transformation goals. China’s economic development has been highly dependent on energy, and the country’s per capita energy use is much higher than the world consumption level. CO2 emissions in China were 9.429 billion tons in 2018, an increase of 2.053 billion tons compared with that in 2008, accounting for 27.8% of global CO2 emissions [10]. As the largest emitter, China is facing pressure concerning both emission abatement and environmental protection [11]. At present, the share of renewable energy is comparatively low in energy utilization structure, which greatly limits the level of innovation and sustainable development within China’s manufacturing industry. Facing the dual dilemmas of an economic development bottleneck and resource constraints, China urgently needs to identify effective technical approaches to the promotion of green innovation to realize sustainability.
How do the enterprises’ economic and innovation activities affect the green innovation efficiency and sustainable economy? And whether the high-quality development transformation of the urban agglomeration region has been paying off? Based on these urgent to be solved problems, one of the most important purpose of our study is to scientifically assess the green innovation efficiency and the quantitative identification of its socioeconomic factors behind in the Pearl River Delta (PRD) urban agglomeration, we constructed an evaluation index system. Referring to existing literature, the concept of “green innovation efficiency” is generally recognized as a sophisticated innovation efficiency which can promote the coordinated growth of “economic-ecological-social system” [14-17]. Green innovation efficiency is always calculated as a ratio considering economic and environmental input and output of ogranization’s innovation activities[18]. It is a low-carbon index of innovation and environmental pollution that indicates the contribution of unit green innovation input to the output. Using a super slacks-based measure (Super-SBM) model, we evaluate the green innovation efficiency of the PRD cities, as well as comparing the efficiency of each city horizontally and analyzing the change trend vertically within the study period. The socioeconomic factors of green innovation efficiency in the PRD urban agglomeration are analyzed by Tobit panel regression model. And we ultimately putting forward a series of policy suggestions to help achieve the goals of industrial transformation and an improved economic development quality.
Point 3: Literature review section: The part from the lines: 118-145 should be shortened and move to the Methodology and data section.
Response 3: Thanks very much for your comments. According to your suggestions, we have shortened contents of lines 118-145, and integrate this part into the Methodology and data section. The revise manuscript are as follows:
- Methodology and data
3.1 The Super-SBM model
The frontier analysis method is widely used by researchers to measure technological efficiency [38]. Frontier analysis consists of two basic analysis methods: stochastic frontier analysis (SFA) and data envelopment analysis (DEA) [39]. In the process of evaluating innovation efficiency, model selection and improvement influence both data acquisition and result requirements, and also the capacity to modify the model by introducing other parameters. For measuring green innovation efficiency, two approaches can be adopted to construct an evaluation system. The first is to set criteria and dimensions in relation to green innovation efficiency, select an indicator system, and use the entropy method for giving the weight to variables [40]; the other is to use an SFA method based on an input-output perspective [41]. Applying these methods, a number of scholars have explored the relationship between technology and environmental efficiency [42]. And many earlier findings also emphasize the importance of policy limitations with regard to CO2 emission reduction [43].
Point 4: Methodology and data section: a really good part of the paper.
Response 4: Thank you very much for your recognition and appreciation of our work.
Point 5: Results and Discussion section: You should interpret and describe the significance of the findings in light of what was already known about the research problem being investigated and to explain any new understanding or insights that emerged as a result of your study of the problem. The discussion should always connect to the introduction by way of the research questions or hypotheses you posed.
Response 5: Thank you very much for your suggestions. According to your suggestions, we have carefully revised and polished the part of discussion and conclusions. Based on the question posed in the introduction, we strengthened the logic connection between the research purpose and the result demonstration. You can review the corresponding revision in the revised manuscript, and the main revised contents as follows:
“From 2010 to 2011, Zhuhai, Shaoguan, Zhaoqing, Qingyuan and Yunfu were in the state of technical ineffectiveness among the PRD cities, which was less than that in 2009. While the polarization of green innovation efficiency is still obvious. Among these five cities, the ineffective state of Zhuhai is due to the low scale efficiency. The others are mainly constrained by their obviously low pure technical efficiency, indi-cating that the utilization efficiency of innovative manpower and capital in these cities is urgent need to improve. Meanwhile, through the value of cities’ scale efficiency, it reveals that government and enterprises still need attach importance to the investment scale of green innovation resources in the PRD. Significantly, Foshan's ineffective status changed that from pure technical efficiency to scale efficiency in 2014-2015, indicating that the city's resource management and allocation level has been notably improved. As the investment in green innovation resources promotes the rapid development of green innovation growth.”
“The green innovation of the PRD cities is still in the initial development stage. Compared with independent research and development, relevant enterprises and in-dustries are more inclined to purchase and introduce mature and systematic technol-ogy from abroad. Besides, they prefer to invest capital, human and other resources in the use of foreign technology, as well as strengthen environmental regulation, super-vision and management. Thus, it results in the non-synchronization of technical utili-zation and technological progress. To sum up, governments and enterprises of the PRD could transform less capital, labor, and energy into higher quality green innovation outputs by decreasing green innovation resources inputs and reducing the negative environmental effects in the process of green innovation activities. The decreases wit-nessed in the values of the technical progress index indicate that room for development exists in relation to the research and development abilities of cities in the PRD, in the process of pursuing green innovation behaviors through adjustments to inputs and outputs.”
Point 6: Conclusions section: I have no comments on this part.
In conclusion, the article requires some revisions before being published.
Response 6: Thank you very much for your recognition and appreciation of our work.
Once again, we appreciate sincerely for the reviewers’ warm and patient work, and thank you very much for your constructive comments and suggestions which really promoted the quality of our manuscript. We hope that our earnest revision will conform to the approval of International Journal of Environmental Research and Public Health.
*********************************************************************
References
Berkel Rv. Eco-Innovation: opportunities for advancing waste prevention. International Journal of Environmental Technology and Management. 2007;7:527-50.
Chen YS, Lai SB, Wen CT. The Influence of Green Innovation Performance on Corporate Advantage in Taiwan. Journal of Business Ethics. 2006;67:p.331-9.
Oltra V, Jean MS. Sectoral Systems Of Environmental Innovation: An Application To The French Automotive Industry. Technological Forecasting & Social Change. 2009;76:567-83.
Publishing O. OECD Green Growth Studies Towards Green Growth: Monitoring Progress: OECD Indicators. Sourceoecd Environment & Sustainable Development. 2011:i-146.
Fan F, Lian H, Liu X, Wang X. Can environmental regulation promote urban green innovation Efficiency? An empirical study based on Chinese cities. Journal of Cleaner Production. 2021;287.
- Statistical Review of World Energy. In: p.l.c. B, editor. UK: BP; 2019. p. 1-68.
IEA. CO2 Emissions from Fuel Combustion 2019 Highlights. Paris: IEA; 2019.
Huang J, Dua D, Hao Y. The driving forces of the change in China's energy intensity: An empirical research using DEA-Malmquist and spatial panel estimations. Economic Modelling. 2017;65:41-50.
Tone K. A slacks-based measure of super-efficiency in data envelopment analysis. European Journal of Operational Research. 2002;143:32-41.
Oh DH. A global Malmquist-Luenberger productivity index. Journal of Productivity Analysis. 2010;34:183-97.
Caves DW, Christensen LR, Diewert WE. Multilateral Comparisons of Output, Input, and Productivity Using Superlative Index. The Economic Journal. 1982;92:73-86.
Chung Y, Fare R, Grosskopf S. Productivity and Undesirable Outputs: A Directional Distance Function Approach. Journal of Environmental Management. 1997;51:229-40.
Kortelainen M. Dynamic environmental performance analysis: A Malmquist index approach. Ecological Economics. 2008;64:p.701-15.
Zhou P, Ang BW, Han JY. Total factor carbon emission performance: A Malmquist index analysis. Energy Economics. 2010;32:194-201.
Shuai S, Fan Z. Modeling the role of environmental regulations in regional green economy efficiency of China: Empirical evidence from super efficiency DEA-Tobit model. J Environ Manage. 2020;261:110227.
Zeng L, Lu H, Liu Y, Zhou Y, Hu H. Analysis of Regional Differences and Influencing Factors on China’s Carbon Emission Efficiency in 2005–2015. Energies. 2019;12:3081-102.
Takeshi A. The Estimation of a Simultaneous-Equation Tobit Model. International Economic Review. 1979;20:169-81.
Ahn H, L.Powell J. Semiparametric estimation of censored selection models with a nonparametric selection mechanism. Journal of Econometrics. 1993;58:3-29.
Cuesta RA, Lovell CAK, Zofio JL. Environmental efficiency measurement with translog distance functions: A parametric approach. Ecological Economics. 2009;68:2232-42.
Hattori T. Relative Performance of U.S. and Japanese Electricity Distribution: An Application of Stochastic Frontier Analysis. Journal of Productivity Analysis. 2003;19:115-.
Huang J, Yu Y, Ma C. Energy Efficiency Convergence in China: Catch-Up, Lock-In and Regulatory Uniformity. Environ Resource Econ. 2018;70:107-30.
Lin B, Du K. Technology gap and China's regional energy efficiency: A parametric metafrontier approach. Energy Economics. 2013;40:529-36.
Zhou Z, Liu C, Zeng X, Jiang Y, Liu W. Carbon emission performance evaluation and allocation in Chinese cities. Journal of Cleaner Production. 2018;172:1254-72.
Cheng Z, Li W. Independent R and D, Technology Introduction, and Green Growth in China’s Manufacturing. Sustainability. 2018;10.

Round 2
Reviewer 1 Report
I don't have any further comments. The paper is suitable for publication. The authors dealt with all my equiries. Thank you and good luck with your future research. I look forward to reading your work in the near future!
Reviewer 2 Report
The authors have comprehensively addressed all comments. The manuscript should be published in its current form.